# Cellular and molecular signatures of in vivo imaging measures of GABAergic neurotransmission in the human brain

Paulina Barbara Lukow [1✉], Daniel Martins [2,3], Mattia Veronese [2,3,4], Anthony Christopher Vernon [5,6], Philip McGuire [1,3], Federico Edoardo Turkheimer [2] & Gemma Modinos [1,2,6]

Diverse GABAergic interneuron networks orchestrate information processing in the brain. Understanding the principles underlying the organisation of this system in the human brain, and whether these principles are reflected by available non-invasive in vivo neuroimaging methods, is crucial for the study of GABAergic neurotransmission. Here, we use human gene expression data and state-of-the-art imaging transcriptomics to uncover co-expression patterns between genes encoding GABA$_A$ receptor subunits and inhibitory interneuron subtype-specific markers, and their association with binding patterns of the gold-standard GABA PET radiotracers [$^{11}$C]Ro15-4513 and [$^{11}$C]flumazenil. We found that the inhibitory interneuron marker somatostatin covaries with GABA$_A$ receptor-subunit genes *GABRA5* and *GABRA2*, and that their distribution followed [$^{11}$C]Ro15-4513 binding. In contrast, the inhibitory interneuron marker parvalbumin covaried with GABA$_A$ receptor-subunit genes *GABRA1*, *GABRB2* and *GABRG2*, and their distribution tracked [$^{11}$C]flumazenil binding. Our findings indicate that existing PET radiotracers may provide complementary information about key components of the GABAergic system.

[1] Department of Psychosis Studies, Institute of Psychiatry, Psychology & Neuroscience, King's College London, 16 De Crespigny Park, SE5 8AF London, UK. [2] Department of Neuroimaging, Institute of Psychiatry, Psychology & Neuroscience, King's College London, 16 De Crespigny Park, SE5 8AF London, UK. [3] NIHR Maudsley Biomedical Research Centre, De Crespigny Park, Denmark Hill, London SE5 8AF, UK. [4] Department of Information Engineering, University of Padua, Via Giovanni Gradenigo, 6, 35131 Padova, PD, Italy. [5] Department of Basic & Clinical Neuroscience, Maurice Wohl Clinical Neuroscience Institute, 5 Cutcombe Road, Brixton, London SE5 9RT, UK. [6] MRC Centre for Neurodevelopmental Disorders, King's College London, New Hunt's House, Guy's Campus, London, UK. ✉email: paulina.lukow@kcl.ac.uk

Whilst accounting for <30% of cortical cells, GABAergic inhibitory interneurons control information processing throughout the brain[1–4]. Their diverse functions include input gating into cortical[5] and subcortical[6] structures, regulating critical period boundaries, homoeostasis[7] and local network activity, as well as entraining cortical network oscillations[1]. Due to their critical role in such wide range of brain functions, inhibitory interneuron dysfunction has been robustly implicated in several psychiatric and neurological conditions, including affective disorders[6,8,9] and schizophrenia[7,10,11].

The GABAergic system comprises diverse inhibitory interneuron subtypes, innervating different excitatory and inhibitory neural targets through a variety of receptors[1]. Inhibitory interneurons vary in their firing threshold, spiking frequency and location of postsynaptic cell innervation, which makes them fit for various functions including the control of synaptic input into the local network and neuronal output regulation[5,12]. This multitude of inhibitory interneurons can be classified through the expression of specific proteins (markers)[13]. While the vast majority of inhibitory interneurons are positive for either parvalbumin (PVALB), somatostatin (SST), or vasoactive intestinal peptide (VIP), further specific subtypes can be identified through the expression of other markers, such as cholecystokinin (CCK)[3]. This array of neurons achieves fine-tuned inhibitory responses via the ionotropic GABA$_A$ receptor (GABA$_A$R), which mediates the hyperpolarisation of the postsynaptic excitatory or inhibitory target cell. The GABA$_A$R is a pentameric chloride channel which most commonly comprises two α, two β and one γ subunit[14]. There are five subtypes of the α subunit and three each of the β and γ subunits; moreover, β can be replaced by a θ subunit, and γ can be replaced by δ, ε or π[14]. This generates a large variety of receptors, the biology and pharmacology of which are determined by their subunit composition. For instance, the low-affinity α1 subunit-containing GABA$_A$R (GABA$_A$Rα1) mediates phasic or activity-dependent inhibition on the postsynaptic cell, whereas GABA$_A$Rα5 presents higher affinity to GABA, maintaining a more continuous inhibitory tone extrasynaptically[15–19]. This intricacy of the GABAergic system poses a challenge to the investigation of the human GABAergic system in vivo that can be both selective and non-invasive.

Given this complexity, the investigation of the roles that GABAergic neurotransmission may play in human brain function in health and disease requires: (1) knowledge of the basic principles underlying the organization of this intricate system; and (2) the ability to distinguish between the contribution of distinct components of the GABAergic system to in vivo measurements of GABAergic neurotransmission. In this context, positron emission tomography (PET) enables quantification of GABA$_A$R binding in vivo in anatomically defined brain regions. This is achieved through the use of radiolabelled tracers (radiotracers), mainly [11C]Ro15-4513, with high affinity to GABA$_A$Rα5, and [11C]flumazenil, a benzodiazepine site-specific radiotracer with more general affinity to GABA$_A$Rα1-3 and α5[20,21]. Although receptor affinity for these radiotracers has been confirmed in preclinical research[22], it is unknown whether this holds inter-species reliability and whether the distribution of distinct cellular and molecular components of the GABAergic system are reflected in radiotracer binding. Answering these questions would advance our understanding of which inhibitory interneurons and GABA$_A$R sub-types may contribute the most to available neuroimaging-based GABA PET measurements. Interestingly, both the distribution of inhibitory interneurons and GABA PET radiotracer binding are heterogeneous across the human brain. For instance, SST and PVALB follow an anticorrelated distribution[11], as do the binding patterns of [11C]Ro15-4513 and [11C]flumazenil[23]. Moreover, there is abundant rodent evidence

of an association between the expression of specific GABA$_A$R subunits, encoded by individual genes in both presynaptic and target neurons (excitatory or inhibitory), and specific inhibitory interneuron subtypes[19,24–26]. However, whether analogous organisation patterns are present in human is unclear. Brain-wide gene expression atlases such as the Allen Human Brain Atlas (AHBA) are increasingly being used to address these questions[27]. The approach involves testing for covariance between the spatial expression pattern of each gene in the AHBA and the spatial topography of neuroimaging measures. This results in the identification of specific cellular and molecular candidates which might contribute the most to the spatial variability in the neuroimaging signal. Hence, using this approach for GABA PET radiotracers will inform how GABAergic system organisation in the human brain may be captured by these existing neuroimaging measures.

Here, we sought to address this issue by using state-of-the-art imaging transcriptomics to (1) uncover patterns of co-expression between genes encoding GABA$_A$R subunits and inhibitory interneuron markers in the human brain, and to (2) decode their links to the binding distribution patterns of two gold-standard GABA PET radiotracers, [11C]Ro15-4513 and [11C]flumazenil[28]. We found that distinct GABAergic inhibitory interneuron marker genes co-expressed with specific GABA$_A$R subunit-encoding genes. Furthermore, we observed that a substantial portion of the variation in [11C]Ro15-4513 and [11C]flumazenil binding covaried with the expression of genes encoding specific components of the GABAergic system. Our findings provide a framework to bridge the gap between genes, cells and macroscopic molecular neuroimaging features of the GABAergic system that may be of help for future in vivo studies in health and disease.

## Results

**GABAergic inhibitory interneuron markers co-express with specific GABA$_A$R subunit-encoding genes.** Our first aim was to uncover patterns of co-expression between genes encoding GABA$_A$R subunits and inhibitory interneuron markers. The latter comprised: the GABA-synthesising enzymes GAD67 (GAD1) and GAD65 (GAD2)[29], parvalbumin (PVALB)[25], somatostatin (SST)[30], vasoactive intestinal peptide (VIP)[31], cholecystokinin (CCK), neuropeptide Y (NPY)[32], calbindin (CALB1)[30], calretinin (CALB2)[30], neuronal nitric oxide synthase (NOS1)[3], reelin (RELN)[33], and the tachykinin precursor genes TAC1, TAC3 and TAC4[34], selected according to preclinical literature and Petilla classification of GABAergic inhibitory interneurons[13]. Data on all available GABA$_A$R subunit genes passing quality threshold were included: α1-5 (GABRA1-5), β1-3 (GABRB1-3), γ1-3 (GABRG1-3), ε (GABRE) and δ (GABRD). Genes encoding subunits α6 (GABRA6), π (GABRP), θ (GABRQ) and ρ1-3 (GABRR1-3) were not used in further analyses as they did not show levels of expression above background. We performed weighted gene co-expression network analysis (WGCNA)[35] on gene expression data from the AHBA[36]. This dataset contained microarray data on 15,633 genes from six post-mortem samples across the left (n = 6 healthy donors) and right hemispheres (n = 2 healthy donors), which were resampled into 83 brain regions of the Desikan–Killiany atlas[37]. WGCNA is a data-driven approach that allows to identify clusters (modules) of highly correlated genes across the whole transcriptome[35].

WGCNA identified 52 co-expression clusters, 13 of which included genes encoding inhibitory interneuron markers and GABA$_A$R subunits of interest. We selected these 13 clusters to investigate which genes shared cluster allocation (Fig. 1a). SST was located in the same cluster as GABRA5, GABRA2 and GABRB1. PVALB had its own cluster (i.e., it was not located in

the same cluster as any other gene of interest). *VIP* was found in the same cluster as *CCK* and no other genes of interest. As those three individual clusters included three of the main non-overlapping inhibitory interneuron markers, labelling the majority of GABAergic inhibitory cells in the mammalian brain[2], we investigated their enrichment in genes co-expressed in specific cell types defined by previous single-cell transcriptomic analysis[38] using the WEB-based GEne SeT AnaLysis Toolkit[39]. These analyses revealed inhibitory interneuron cell-type enrichment in the *SST* and *PVALB* clusters, and excitatory cell-type enrichment in the *VIP* cluster (Supplementary Fig. 1). This indicated that the subsequent analyses including the *SST* and *PVALB* clusters were relevant for networks involving GABAergic inhibitory interneuron cell-types. Other separate clusters of genes included *GAD1, GABRA1, GABRB2, GABRG2* and *GABRG3; CALB1, CALB2, GABRG1* and *GABRE; GABRA4, NPY* and *TAC3;* and *GAD2* and *NOS1*. Finally, *GABRB1, GABRB3, GABRA3, RELN, TAC1, GABRD* and *TAC4* were all found individually in separate clusters that did not share assignment with any other gene of interest.

The WGCNA cluster-based findings were complemented by a pairwise correlation analysis. This served both as a validation step and as a method to investigate co-expression patterns between genes of interest that might not pertain to a discrete WGCNA cluster. Hence, we performed bivariate correlation analysis of the genes of interest with the *corrplot* package in R 4.0.3 (Fig. 1b). This revealed strong correlations (Pearson's $r > 0.5$, $p < 0.05$) between *SST* and *GABRA5, GABRA2* and *GABRB1* (Fig. 1c–e); between *VIP* and *CCK* (Fig. 1f); between *GABRA1* and *GABRB2, GABRG2* and *GABRG3* (Fig. 1h–j); and between *PVALB* and genes encoding the subunits of the main GABAergic receptor in the brain, GABA$_A$Rα1β2γ2 (*GABRA1, GABRB2* and *GABRG2*)[19] (Fig. 1k–m).

**[11C]Ro15-4513 and [11C]flumazenil PET binding track the expression of specific genes encoding GABAergic inhibitory interneuron markers and GABA$_A$R subunits**. Our second aim was to decode the links between the co-expression patterns of genes encoding GABA$_A$R subunits and inhibitory interneuron markers (identified through the first aim) and the binding patterns of two gold-standard GABA PET radiotracers, [11C]Ro15-4513 and [11C]flumazenil. For this purpose, we integrated the gene expression data with maps of [11C]Ro15-4513 binding ($n = 10$ healthy volunteers) and [11C]flumazenil binding ($n = 16$ healthy volunteers) through a covariance analysis. We used partial least square (PLS) regression, accounting for spatial autocorrelation. For each radiotracer, we performed two complementary analyses. First, we used as predictors the eigengenes of each of the clusters we identified in the WGCNA analysis. Eigengenes in this context refer to the first principal component of a given cluster, thus representing the pattern of regional expression of all genes within that cluster. Second, we used as predictors all 15,633 genes that passed our pre-processing criteria and inspected the rank of each of our genes of interest in the ranked list of genes according to their spatial alignment with the radiotracer. This would provide a sense of how specific the correlation of each of our genes of interest might be as compared to other non-hypothesized genes and the cluster-wise analysis.

**[11C]Ro15-4513 binding is associated with *SST, GABRA5, GABRA2* and *GABRB1* expression**. For [11C]Ro15-4513, the first PLS component (PLS1) of the cluster-wise analysis explained alone the largest amount (58.28%, $p_{spatial} < 0.0001$) of variance in radiotracer binding (Supplementary Fig. 4a). We focused our subsequent analyses on this first component, as it explained the

most of variance. The cluster containing *GABRA5, GABRA2, SST* and *GABRB1* was assigned the highest positive PLS1 weight ($Z = 6.18$, FDR $= 1.67 \times 10^{-8}$). In the gene-wise PLS analysis, the first PLS component explained alone the largest amount of variance (57.78%, $p_{spatial} < 0.0001$) (Supplementary Fig. 4b). *GABRA5, GABRB3, GABRA2, NPY, VIP, SST, GABRB1, TAC3, CCK, GABRA3, RELN* and *TAC1* ($Z = 4.38–2.28$, $p_{FDR} = 0.000273$-$0.0366$) were all assigned significant positive weights in descending order (Table 1). Interestingly, *PVALB* expression had a significant negative PLS1 weight ($Z = -2.46$, $p_{FDR} = 0.0255$), which suggested a negative relationship between *PVALB* expression and [11C]Ro15-4513 binding. For full PLS results, see Supplementary Data 1 and 2.

The radiotracer binding and the distribution of weights resulting from both PLS analyses (cluster-wise and gene-wise) followed and antero-posterior distribution gradient in the brain (Fig. 2), consistent with the analogous gradient of *SST* expression shown previously[11]. We then followed up these results with a cell-type enrichment analysis, accounting for the weights associated with each gene included in the analysis. This analysis revealed enrichment in genes expressed in SST, CCK and VIP/CCK inhibitory interneurons (Supplementary Fig. 2). The result supported an association between the distribution of these cell types with [11C]Ro15-4513 binding.

**[11C]flumazenil binding is associated with *PVALB, GABRA1, GABRB2, GABRG2, GABRG3* and *GAD1* expression**. For [11C]flumazenil, the first PLS component (PLS1) of the cluster-wise analysis explained alone the largest amount of variance (36.49%, $p_{spatial} = 0.001$) in radiotracer binding (Supplementary Fig. 4c). The cluster containing *GABRB2, GABRG3, GABRA1, GABRG2* and *GAD1* was assigned the highest positive PLS1 weight ($Z = 7.48$, $p_{FDR} = 1.87 \times 10^{-12}$). In the gene-wise PLS analysis, the first PLS component explained alone the largest amount of variance (37.13%, $p_{spatial} = 0.005$) (Supplementary Fig. 4d). *GABRB2, GABRD, GABRG3, GABRA1, GABRG2, GABRA4, GAD1, VIP, CCK* and *PVALB* ($Z = 6.91–2.26$, $p_{FDR} = 1.45 \times 10^{-9}$-$0.0315$) were all assigned significant positive weights in descending order (Table 2). For full PLS results, see Supplementary Data 3 and 4

Radiotracer binding, as well as the distribution of weights resulting from both PLS analyses, followed a postero–anterior distribution gradient (Fig. 3), consistent with analogous pattern of *PVALB* expression shown previously[11]. Following up these results with a cell-type enrichment analysis, accounting for weights associated with each gene input into the analysis, revealed enrichment in genes expressed in PVALB, CCK, VIP/CCK and SST inhibitory interneurons (Supplementary Fig. 3). The result supported an association between the distribution of PVALB, CCK and VIP/CCK cell-types with [11C]flumazenil binding, and suggested an association between genes enriched in the SST cell-type and radiotracer signal despite no direct covariance with *SST* expression found in the PLS analysis. Interestingly, [11C]flumazenil binding was negatively associated with astrocyte-enriched genes.

## Discussion

Our main finding was that the spatial pattern of expression of specific inhibitory interneuron marker genes covaried with that of different GABA$_A$R subunit genes and that these co-expression patterns explained a substantial portion of the variation in GABA PET radiotracer binding, a measure of in vivo neurotransmission. Specifically, while [11C]Ro15-4513 binding followed an anterior distribution that tracked the expression of *GABRA5* and *SST*, [11C]flumazenil followed a more posterior distribution which

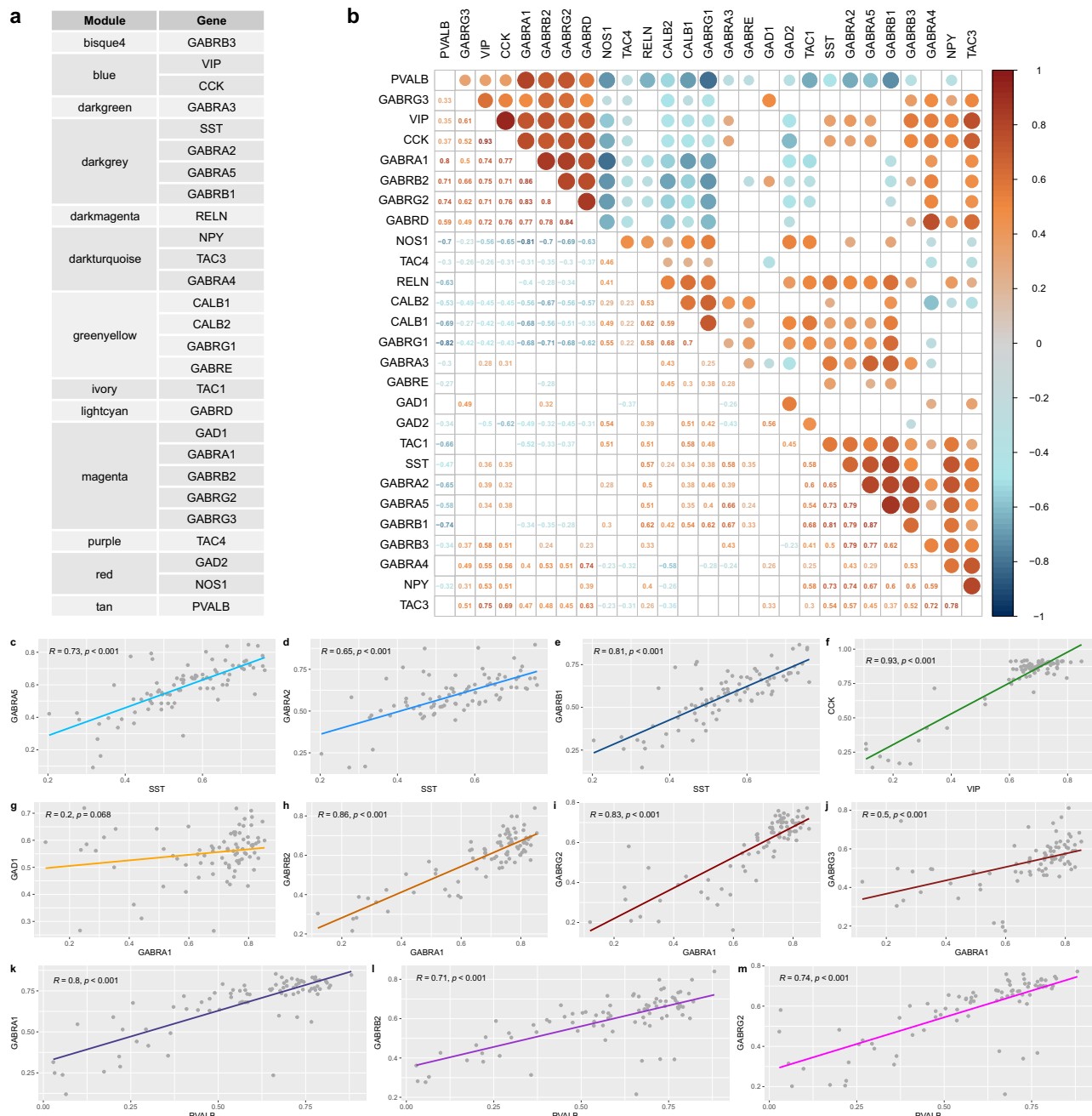

**Fig. 1 Specific GABAergic interneuron markers co-express with different GABA$_A$R subunits. a** Co-expression cluster assignment and **b** bivariate correlations ($p < 0.05$) between GABAergic interneuron markers and GABA$_A$R subunits. Pairwise correlations between **c–e** somatostatin (*SST*), **f** vasoactive intestinal peptide (*VIP*), **g–j** *GABRA1* and **k–m** parvalbumin (*PVALB*), and other genes of interest sharing their cluster assignment in the AHBA dataset. *CALB1*, calbindin, *CALB2*, calretinin, *CCK*, cholecystokinin, *GABRA1-5*, GABA$_A$R receptor subunits α1-5, *GABRB1-3*, GABA$_A$R receptor subunits β1-3, *GABRD*, GABA$_A$R receptor subunit δ, *GABRE*, GABA$_A$R receptor subunit ε, *GABRG1-3*, GABA$_A$R receptor subunits γ1-3, *GAD1*, GABA-synthesising enzyme GAD67, *GAD2*, GABA-synthesising enzyme GAD65, *NOS1*, neuronal nitric oxide synthase, *NPY*, neuropeptide Y, *PVALB*, parvalbumin, *RELN*, reelin, *SST*, somatostatin, *TAC1*, *TAC3* and *TAC4*, the tachykinin precursor genes, *VIP*, vasoactive intestinal peptide.

covaried with the expression of *GABRA1* and *PVALB*. Previous PET studies described that [$^{11}$C]Ro15-4513 and [$^{11}$C]flumazenil binding were anticorrelated along an anterior-posterior axis[23], convergent with recent findings of a largely developmentally preserved gradient of *SST* to *PVALB* distribution in the human brain[11]. Our investigation suggests that these findings may not be coincidental and that these two GABA PET radiotracers could provide complementary information about the architecture of the inhibitory system. These findings may have important

implications for the study of GABAergic dysfunction in neuropsychiatric conditions, as discussed below.

Preclinical studies have shown that GABA$_A$Rα5 are enriched on principal cell membranes targeted by SST cells[5,26,40]. Consistent with these findings, we observed that the spatial pattern of [$^{11}$C]Ro15-4513 binding covaried most strongly with gene expression from a cluster containing *GABRA5*, *GABRA2* and *SST*. In previous research, [$^{11}$C]Ro15-4513 was shown to present 10–15-fold higher affinity to human cloned GABA$_A$Rα5 than to

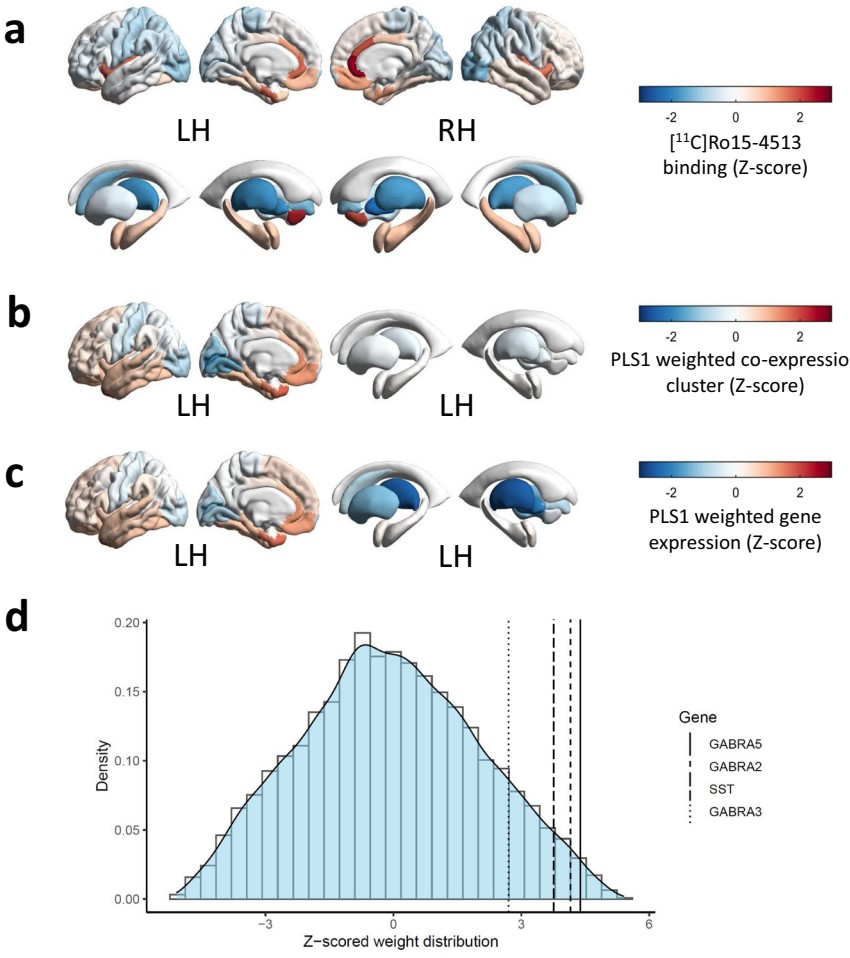

**Fig. 2 [¹¹C]Ro15-4513 binding follows an antero-posterior gradient and spatially tracks *SST*, *GABRA5*, *GABRA2* and *GABRA3* expression.** Z-scored regional brain distribution of **a** [¹¹C]Ro15-4513 binding, **b** weights of covariance between [¹¹C]Ro15-4513 signal and 52 co-expression clusters from the AHBA and **c** weights of covariance between [¹¹C]Ro15-4513 signal and expression of genes from the AHBA. **d** Density plot of Z-scored weight distribution of genes from the AHBA in their covariance with [¹¹C]Ro15−4513 signal, with location of *GABRA5*, *GABRA2*, *GABRA3* and *SST*. *GABRA2*, GABA$_A$ receptor subunit α2, *GABRA3*, GABA$_A$ receptor subunit α3, *GABRA5*, GABA$_A$ receptor subunit α5, *SST*, somatostatin.

**Table 1 Weights and significance of covariance between the expression of individual genes of interest and [¹¹C]Ro15-4513 signal.**

| Gene | PLS rank / 15,633 | PLS gene weight (Z-score) | $p_{FDR}$ |
|---|---|---|---|
| *GABRA5* | 183 | 4.38 | $2.73 \times 10^{-4}$ |
| *GABRB3* | 275 | 4.20 | $4.11 \times 10^{-4}$ |
| *GABRA2* | 306 | 4.15 | $4.65 \times 10^{-4}$ |
| *NPY* | 523 | 3.83 | $1.02 \times 10^{-3}$ |
| *VIP* | 527 | 3.82 | $1.02 \times 10^{-3}$ |
| *SST* | 573 | 3.76 | $1.19 \times 10^{-3}$ |
| *GABRB1* | 804 | 3.48 | $2.41 \times 10^{-3}$ |
| *TAC3* | 954 | 3.31 | $3.69 \times 10^{-3}$ |
| *CCK* | 1492 | 2.84 | 0.0114 |
| *GABRA3* | 1665 | 2.71 | 0.0153 |
| *RELN* | 2066 | 2.43 | 0.0271 |
| *TAC1* | 2283 | 2.28 | 0.0366 |
| *PVALB* | 13,433 | −2.46 | 0.0255 |

Statistically significant results ($p_{FDR}$ < 0.05) shown only. PLS weight and $p_{FDR}$ shown to third significant figure. PLS, partial least squares regression analysis. *CCK*, cholecystokinin, *GABRA2−5*, GABA$_A$R receptor subunits α2-5, *GABRB1/3*, GABA$_A$R receptor subunits β1/3, *NPY*, neuropeptide Y, *PVALB*, parvalbumin, *RELN*, reelin, *SST*, somatostatin, *TAC1* and *TAC3*, tachykinin precursor genes, *VIP*, vasoactive intestinal peptide.

GABA$_A$Rα1-3[23], and the co-expression of *GABRA2* and *GABRA5* was shown by immunohistochemistry in the rat brain[16], also in line with our observations. Interestingly, we also found covariance between the expression of *CCK* and *GABRA3* and the distribution of [¹¹C]Ro15-4513 signal. GABA$_A$Rα2/3 expression in rodents has been most consistently found in the post-synapse of principal cells targeted by CCK basket cells[3,19,24,41]. It is plausible that the association we found between [¹¹C]Ro15-4513 binding and *CCK*, *GABRA2* and *GABRA3* expression is circumstantial, if enough spatial overlap between the expression of these genes with *SST* and *GABRA5* exists. Both *GABRA2*[41] and *GABRA3*[42] are highly expressed in the hippocampus, where α5 is most enriched[14]. Moreover, *GABRA2* and *GABRA5* follow a similar expression pattern in several brain regions[42]. Alternatively, the finding might reflect secondary affinity of the radiotracer to GABA$_A$Rα2/3. While [¹¹C]Ro15-4513 has 10-15-times greater affinity towards GABA$_A$Rα5 than to GABA$_A$Rα1-3[23], it is possible that in regions where GABA$_A$Rα5 is lower, this radiotracer binds secondarily to GABA$_A$Rα2/3. This might deserve further investigation, especially as GABA$_A$Rα5 constitute <5% of all GABA$_A$Rs in the brain[41] and PET radiotracers are administered systemically with an intravenous injection, which allows for secondary binding to occur. However, our approach relied on the interrogation of indirect spatial associations between PET radiotracer binding and gene

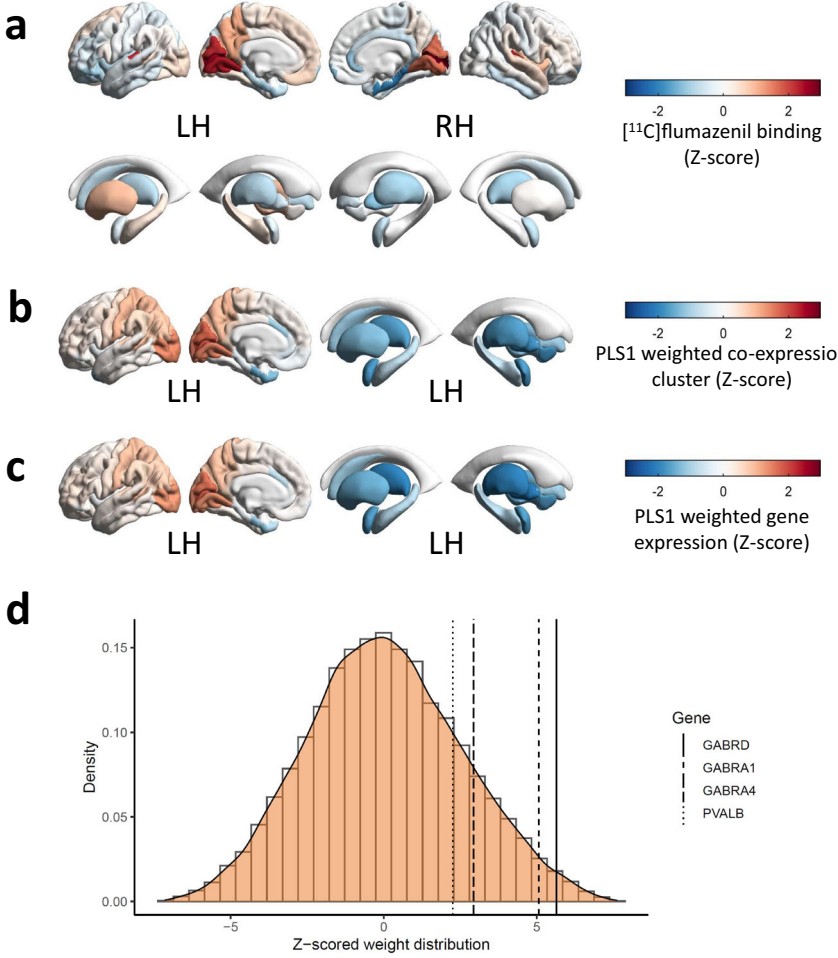

**Fig. 3 [¹¹C]flumazenil binding follows a postero–anterior gradient and spatially tracks _PVALB_, _GABRD_, _GABRA1_ and _GABRA4_ expression.** Z-scored regional brain distribution of **a** [¹¹C]flumazenil binding, **b** weights of covariance between [¹¹C]flumazenil signal and 52 co-expression clusters from the AHBA and **c** weights of covariance between [¹¹C]flumazenil signal and expression of genes from the AHBA. **d** Density plot of Z-scored weight distribution of genes from the AHBA in their covariance with [¹¹C]flumazenil signal, with location of _GABRD_, _GABRA1_, _GABRA4_ and _PVALB_. _GABRA1_, GABA_A receptor subunit α1, _GABRA4_, GABA_A receptor subunit α4, _GABRD_, GABA_A receptor subunit δ, _PVALB_, parvalbumin.

**Table 2 Weights and significance of covariance between the expression of individual genes of interest and [¹¹C] flumazenil signal.**

| Gene | PLS rank/ 15,633 | PLS gene weight (Z-score) | $p_{FDR}$ |
|---|---|---|---|
| _GABRB2_ | 22 | 6.91 | $1.45 \times 10^{-9}$ |
| _GABRD_ | 227 | 5.64 | $3.95 \times 10^{-7}$ |
| _GABRG3_ | 392 | 5.11 | $3.97 \times 10^{-6}$ |
| _GABRA1_ | 401 | 5.07 | $4.67 \times 10^{-6}$ |
| _GABRG2_ | 597 | 4.67 | $2.31 \times 10^{-5}$ |
| _GABRA4_ | 2101 | 2.94 | $6.43 \times 10^{-3}$ |
| _GAD1_ | 2405 | 2.70 | 0.0117 |
| _VIP_ | 2566 | 2.60 | 0.0150 |
| _CCK_ | 3050 | 2.26 | 0.0311 |
| _PVALB_ | 3059 | 2.26 | 0.0315 |
| _GABRA3_ | 12,482 | −2.09 | 0.0436 |
| _GABRG1_ | 12,816 | −2.29 | 0.0293 |
| _CALB1_ | 13,166 | −2.50 | 0.0190 |
| _NOS1_ | 13,727 | −2.91 | $6.70 \times 10^{-3}$ |
| _CALB2_ | 15,393 | −5.09 | $4.34 \times 10^{-6}$ |

Statistically significant results ($p_{FDR} < 0.05$) shown only. PLS weight and $p_{FDR}$ shown to third significant figure. PLS, partial least squares regression analysis. _CALB1_, calbindin, _CALB2_, calretinin, _CCK_, cholecystokinin, _GABRA1/3/4_, GABA_AR receptor subunits α1/3/4, _GABRB2_, GABA_AR receptor subunit β2, _GABRD_, GABA_AR receptor subunit δ, _GABRG1-3_, GABA_AR receptor subunits γ1-3, _GAD1_, GABA-synthesising enzyme GAD67, _NOS1_, neuronal nitric oxide synthase, _PVALB_, parvalbumin, _VIP_, vasoactive intestinal peptide.

expression across brain regions. As such, direct extrapolations about specific synaptic contributions to our findings warrant future validation with precise molecular methods such as immunocytochemistry or autoradiography with pharmacological blocking. Patterns in regional protein density of specific subunits may help elucidate whether [¹¹C]Ro15-4513 binds to GABA_ARα2/3 in addition to GABA_ARα5. For instance, future studies may assess interlaminar differences in [¹¹C]Ro15-4513 binding, as α2/3 and α5 present different expression patterns across human prefrontal cortex layers[43]. In addition, future research could investigate the proportion of synaptic to extra-synaptic binding of [¹¹C]Ro15-4513 in human neurons, as GABA_ARα5 are predominantly extrasynaptic[17].

The spatial pattern of [¹¹C]flumazenil binding covaried most strongly with the gene expression cluster containing _GAD1_, _GABRA1_, _GABRB2_, _GABRG2_ and _GABRG3_. There is prior evidence that [¹⁸F]flumazenil accumulation across the mouse brain after mutations in α2, α3 and α5 subunits, but not in α1, remained similar to that in wild-type mice[44]. This may suggest that [¹¹C]flumazenil binding to GABA_ARα1 accounts for most of the PET signal. Our finding aligns with the notion that the abundance of GABA_ARα1 may be reflected in the pattern of flumazenil binding[14,23]. Indeed, GABA_ARα₁β₂γ₂ is the most widely expressed GABA_AR[8,20] and the co-expression of

*GABRA1*, *GABRB2* and *GABRG2* are supported by analogous observation in preclinical immunohistochemistry studies[45,46]. Interestingly, we found that [11C]flumazenil signal was also associated with *PVALB* expression, consistent with our observation of a high association between *PVALB* expression with *GABRA1*, *GABRB2* and *GABRG2*, and with preclinical evidence that post-synaptic membranes of principal cells and PVALB inhibitory interneurons targeted by this interneuron subtype are enriched in GABA$_A$Rα1[19,24–26,41]. Covariance of *GABRA4* and *GABRD* expression with [11C]flumazenil binding is consistent with the finding that those subunits are commonly co-expressed in the forebrain, and that the δ subunit associates extra-synaptically with α1, primarily in the cerebellum where [11C]flumazenil uptake is higher than that of [11C]Ro15-4513[23], but also in the cerebrum[47,48]. As mentioned above, further study using more precise molecular methodology is required to validate these findings.

Interestingly, our cell-type enrichment analysis of radiotracer signal covariance with gene expression showed [11C]flumazenil to be negatively associated with astrocyte-enriched genes, whereas [11C]Ro15-4513 binding did not show such association (Supplementary Figs. 2 and 3). Astrocytes express GABA$_A$Rs[49], and those containing α1, α2, β1 and γ1 were shown to be expressed on rat astrocytes[50]. This suggests some GABA PET radiotracer binding may occur on this cell type. We found [11C]Ro15-4513 and [11C]flumazenil binding to be associated with the expression of genes encoding some of these subunits. We also observed a trend-level enrichment of the astrocyte cell-type in the cluster containing *SST*, which covaried most strongly with [11C]Ro15-4513 signal. Astrocyte distribution is higher in the frontal and temporal cortex[51], where [11C]Ro15-4513 binding is also greater. However, the spatial associations we set out to identify through our approach relied on the levels of radiotracer binding and gene expression. The degree of GABA$_A$R expression[52], as well as *Gabra1* and *Gabra5*[53], was shown in rodents to be lower in astrocytes than in neurons. Moreover, the human brain comprises approximately equal numbers of glial and neural cells, and astrocytes represent around 20% of glial cells[54]. This suggests neural GABA$_A$R expression exceeds that of astrocytes. This may explain the negative covariance between [11C]flumazenil signal and astrocyte-enriched genes, as [11C]flumazenil binding is greatest in posterior brain areas, where astrocyte distribution is lower[51]. Correspondingly, it is plausible that the contribution of radiotracer binding to GABA$_A$Rs on astrocytes might be insufficient to considerably drive spatial variation in [11C]Ro15-4513 binding across the brain.

Our study may have important implications for future research into the role of GABAergic dysfunction in neuropsychiatric disease using PET imaging. A substantial portion of the variation in [11C]Ro15-4513 and [11C]flumazenil binding covaried with the expression of genes encoding distinct cellular and molecular components of the GABAergic system. These findings support the use of these existing GABA PET radiotracers to capture separate, albeit complementary, features of GABAergic dysfunction in those disorders. For instance, altered GABAergic function is hypothesised to contribute to the pathophysiology of affective disorders[8]. GABA$_A$Rα2 and GABA$_A$Rα3 agonism is implicated in benzodiazepine-mediated anxiolysis[14,19], whereas GABA$_A$Rα1 agonism is associated with the undesired sedative effects of benzodiazepines[14,19]. Furthermore, SST inhibitory interneuron dysfunction has been implicated in the aetiology of depressive disorders[41,55], and α5 subunit increases have been found *post-mortem* in patients with depression[9]. Interestingly, SST cell disinhibition produced an anxiolytic- and antidepressant-like effect akin to that of benzodiazepines or ketamine in preclinical models[56]. Our finding that *SST*, *GABRA2* and *GABRA3* co-expression was linked

to [11C]Ro15-4513, but not to [11C]flumazenil binding, suggests that [11C]Ro15-4513 may be useful to investigate GABAergic system components implicated in anxiogenesis and depressive symptomatology.

Furthermore, our findings may have important implications for the study of disorders where multiple GABAergic system abnormalities might occur, such as schizophrenia. PVALB inhibitory interneuron loss was reported in hippocampi of patients with schizophrenia by *post-mortem* examination[57]. We found that [11C]flumazenil binding was associated with *PVALB* expression. Two separate [11C]flumazenil PET studies in schizophrenia reported inconsistent effects across multiple cortical regions for patients with different medication status, while one of these two studies did find binding alterations in subcortical regions including the hippocampus[58,59]. SST inhibitory interneuron cell reductions have also been reported in the hippocampus and cortical regions of schizophrenia patients[57,60]. We found that [11C]Ro15-4513 signal covaried with *SST* expression. A recent [11C]Ro15-4513 study in schizophrenia reported binding decreases in antipsychotic medication-free patients compared to healthy controls, limited to the hippocampus[61]. Analogous deficits were not identified in another sample including currently or previously medicated patients[61,62], which may be consistent with preclinical observations that antipsychotic treatment may affect hippocampal [11C]Ro15-4513 binding[63]. Taken together, these findings may suggest that SST interneuron dysfunction in the hippocampus and PVALB interneuron abnormalities in cortical and subcortical regions may play a role in the onset of psychosis. Future GABA PET studies in early psychosis may address this hypothesis by examining PVALB interneuron-mediated inhibition with [11C]flumazenil and SST interneuron-associated inhibition with [11C]Ro15-4513, while carefully considering the effects of antipsychotic medication.

Our study had some limitations. First, we relied on indirect spatial associations between gene expression and radiotracer binding, which alone does not directly imply co-expression in the same cell or in interconnected neurons, nor direct radiotracer binding. The limited resolution of both methods also did not allow us to discern between specific cell-types or synaptic microcircuits. However, we note that our findings are broadly supported by the preclinical literature using more fine-grained methods, which lends support to the plausibility of our imaging transcriptomics findings in humans. We cannot exclude that the use of two different pipelines for the quantification of [11C]flumazenil and [11C]Ro15-4513 might have had an impact on our results. However, the spatial distribution of radiotracer binding to their target, of importance for this study, has been shown with pharmacological blocking studies to be preserved regardless of which particular quantification method is used[64]. This spatial distribution of signal is also expected to be preserved regardless of the normalisation method originally applied to the [11C]flumazenil map, due to the non-selective nature of binding to benzodiazepine-sensitive receptors of both [11C]flumazenil and diazepam[64]. Second, the relationship between mRNA and their product proteins can be affected by post-translational modifications—in those cases, one cannot assume that the density of mRNA is a good proxy for the distribution of the protein[65]. Our study did not aim to determine the molecular binding of the two radiotracers, but rather their global pattern of covariance with genes encoding specific GABAergic signalling components. Thus, our findings are correlational by nature and must be interpreted with caution. Third, the AHBA includes data from six donors only. Samples from the right hemisphere were only collected for two donors, which led us to restrict our analyses to the left hemisphere. Although not a specific limitation of this study, this raises questions about whether this small sample can capture well the principles of organisation of the canonical architecture of gene expression in the

human brain and generalise well. However, it is noteworthy that previous studies found differences in gene expression between the two hemispheres to be mostly negligible[66]. Finally, because we applied an intensity threshold to the microarray dataset to minimise inclusion of unreliable measures of gene expression, we were not able to investigate some genes of interest, including *GABRA6*, *GABRP*, *GABRQ* and *GABRR1-3*, due to the low intensity of signal for these genes in the AHBA dataset as compared to background. Future studies using high sensitivity methods to measure expression of these genes across the whole brain will help to complement our findings in this respect.

In summary, we provide evidence of the regional association between the expression of: (1) *SST*, *GABRA5* and *GABRA2*; (2) *PVALB* and *GABRA1*, *GABRB2* and *GABRG2*; and (3) *VIP* and *CCK* in the human brain. These findings expand our understanding of the canonical transcriptomic architecture of specific GABAergic system components in the human brain. Furthermore, we provide first evidence that the expression of distinct inhibitory interneuron sub-types and specific GABA$_A$R subunits covary with [$^{11}$C]Ro15-4513 and [$^{11}$C]flumazenil binding in a largely non-overlapping manner. While [$^{11}$C]Ro15-4513 signal covaried with the expression of *SST* and genes encoding several major benzodiazepine-sensitive GABA$_A$R subunits implicated in affective functioning (*GABRA5*, *GABRA2* and *GABRA3*), [$^{11}$C]flumazenil tracked *PVALB* and genes encoding subunits comprising the most widely expressed receptor (*GABRA1*, *GABRB2* and *GABRG2*). These findings may have important implications for existing and future PET studies of GABA dysfunction in neuropsychiatric disease. Once corroborated by more direct molecular methods, our work has the potential to inform methodological choices for imaging the GABAergic system, and to help the interpretation of findings within a framework that bridges the gap between genes, cells and macroscopic in vivo molecular neuroimaging features.

## Methods

**The Allen Human Brain Atlas (AHBA) dataset**. The AHBA dataset includes microarray data of gene expression in *post-mortem* brain samples from six healthy donors (one female, mean age ± SD 42.5 ± 13.38, range 24–57)[36]. Detailed information about donor characteristics and dataset generation can be found in the Allen Human Brain Atlas website (https://human.brain-map.org/). In short, the brain (cerebrum including the brainstem) was sampled systematically across the left hemisphere in all six donors, and the right hemisphere in two of the donors. Manual macrodissection was performed on the cerebral and cerebellar cortex, as well as subcortical nuclei, in 50–200 mg increments. Subcortical areas and cerebellar nuclei were sampled with laser microdissection in 36 mm$^2$ increments. RNA was then isolated from these dissections and gene expression was quantified with microarray.

**Gene expression data: pre-processing and spatial mapping**. The approach was similar to that used in previous literature[67,68]. Human gene expression microarray data were extracted from the AHBA with the *abagen* toolbox (https://github.com/netneurolab/abagen)[69] in JupyterLab Notebook through anaconda3 in Python 3.8.5. We mapped AHBA samples to the parcels of the Desikan–Killiany atlas, including 83 brain regions across both brain hemispheres (34 cortical and seven subcortical regions per brain hemisphere, plus brainstem). Genetic probes were reannotated using information provided by Arnatkeviciute et al., 2019[70] instead of the default probe information from the AHBA dataset to exclude probes that cannot be reliably matched to genes. According to the existing guidelines for probe-to-gene mappings and intensity-based filtering[70], the reannotated probes were filtered based on their intensity relative to background noise level; probes with intensity less than background in ≥50% of samples were discarded. A single probe with the highest differential stability, $\Delta S(p)$, was selected for each gene, where differential stability was calculated as[71] (Eq. 1):

$$\triangle_S(p) = \frac{1}{\binom{N}{2}} \sum_{i=1}^{N-1} \sum_{j=i+1}^{N} \rho[B_i(p), B_j(p)], \tag{1}$$

where $\rho$ is Spearman's rank correlation of the expression of a single probe p across regions in two donor brains, Bi and Bj, and $N$ is the total number of donor brains. This procedure retained 15,633 probes, each representing a unique gene.

Next, tissue samples were assigned to brain regions using their corrected MNI coordinates (https://github.com/chrisfilo/alleninf) by finding the nearest region within a radius of 2 mm. To reduce the potential for misassignment, sample-to-

region matching was constrained by hemisphere and cortical/subcortical divisions. If a brain region was not assigned to any sample based on the above procedure, the sample closest to the centroid of that region was selected in order to ensure that all brain regions were assigned a value. Samples assigned to the same brain region were averaged separately for each donor. Gene expression values were then normalized separately for each donor across regions using a robust sigmoid function and rescaled to the unit interval. Scaled expression profiles were finally averaged across donors, resulting in a single matrix with rows corresponding to brain regions and columns corresponding to the retained 15,633 genes.

The genes of interest list included data on all available GABA$_A$R subunits and inhibitory interneuron markers defined according to the Petilla terminology[13] and the existing animal literature. These were: the GABA-synthesising enzymes GAD67 (*GAD1*) and GAD65 (*GAD2*)[29], parvalbumin (*PVALB*)[25], somatostatin (*SST*)[30], vasoactive intestinal peptide (*VIP*)[31], cholecystokinin (*CCK*), neuropeptide Y (*NPY*)[32], calbindin (*CALB1*)[30], calretinin (*CALB2*)[30], neuronal nitric oxide synthase (*NOS1*)[3], reelin (*RELN*)[33], and the tachykinin precursor genes *TAC1*, *TAC3* and *TAC4*[34]. The genes of interest that did not pass this intensity-based thresholding were *GABRA6*, *GABRP*, *GABRQ* and *GABRR1-3*.

**Weighted gene co-expression network analysis (WGCNA)**. Hierarchical clustering of genes by their expression across brain regions was performed with the WGCNA package[35] in R 4.0.3 for the gene expression dataset. The 'signed' WGCNA method was chosen to form clusters (modules) enriched in genes which expression was positively correlated, indicating co-expression[35]. As we aimed to identify the main features of GABAergic system organisation across the human brain, the analysis included the whole-brain microarray dataset. Gene expression correlation matrix was transformed into an adjacency matrix using the soft threshold power of 14. This adjacency matrix contained pairwise correlations between all genes in the dataset, uncorrected for multiple comparisons. The power value was chosen as it was the first value at which the network satisfied the free-scale topology criterion at $R^2 > 0.8$, therefore maximising mean network node connectivity (Supplementary Fig. 5). The adjacency matrix was then transformed into a dissimilarity measure matrix, representing both the expression correlation between pairs of genes as well as the number of the genes they both highly correlated with positively[35]. Finally, average-linkage hierarchical clustering using the dissimilarity measure was performed. Individual modules were identified through the classic 'tree' dendrogram branch cut[72].

**Parametric map of [$^{11}$C]Ro15-4513 binding**. Ten healthy participants (four females, mean age ± SD 25.40 ± 3.20, range 22–30) with no history of psychiatric diagnoses, neurological illness or head trauma with loss of consciousness were scanned with the radiotracer [$^{11}$C]Ro15-4513. Scanning was performed on a Signa$^{TM}$ PET-MR General Electric (3 T) scanner using the MP26 software (01 and 02) at Invicro, A Konica Minolta Company, Imperial College London, UK. The study was approved by the London/Surrey Research Ethics Committee. All subjects provided written informed consent before participation, in accordance with The Declaration of Helsinki. The radiotracer was administered through the dominant antecubital fossa vein in a single bolus injection, administered at the beginning of the scanning session. The mean ± SD amount of radiation administered was 307 MBq ± 71 (range: 173–405 MBq). PET acquisition was performed in 3D list mode for 70 min and binned in the following frames: 15sx10, 60sx3, 120sx5, 300sx11. Attenuation correction was performed with a ZTE sequence (voxel size: 2.4 × 2.4 × 2.4 mm$^3$, field of view = 26.4, 116 slices, TR = 400 ms, TE = 0.016 ms, flip angle = 0.8º). A T1-weighted IR-FSPGR sequence was used for PET image co-registration (voxel size: 1 × 1 × 1 mm$^3$, field of view = 25.6, 200 slices, TR = 6.992 ms, TE = 2.996 ms, TI = 400 ms, flip angle = 11º).

Individual subject images were generated with MIAKAT v3413 in Matlab R2017a. For each subject, an isotropic, skull-stripped IR-FSPGR structural image normalised to the MNI template was co-registered onto an isotropic, motion-corrected integral image created from the PET time series. Binding potential parametric maps were estimated through a simplified reference tissue model using the pons as the reference region and solved with basis function method[73]. The individual parametric maps were averaged using SPM imCalc function and resliced with the Co-register: Reslice function to match the dimensions of the Desikan–Killiany atlas (voxel size 1 × 1 × 1 mm, number of voxels per direction X = 146, Y = 182, Z = 155). Finally, the averaged parametric map of [$^{11}$C]Ro15-4513 binding was resampled into 83 regions of the Desikan–Killiany atlas space using the fslmeans function from FSL.

**Parametric map of [$^{11}$C]flumazenil binding**. An averaged parametric map of maximal binding of the [$^{11}$C]flumazenil ([$^{11}$C]Ro15-1788) radiotracer was downloaded from an open-access dataset made available by the Neurobiology Research Unit at Copenhagen University Hospital (https://xtra.nru.dk/BZR-atlas/). In brief, 16 healthy participants between 16–46 years old (nine females, mean age ± SD 26.6 ± 8) were scanned on a CTI/Siemens High-Resolution Research Tomograph. Distribution volume parametric maps were obtained from blood-based Logan graphical analysis and then normalised to *post-mortem* human brain [$^3$H]diazepam autoradiography data for absolute quantification of benzodiazepine-sensitive

receptor density. For full details on the generation of this map, please refer to the original publication[74].

**Statistics and reproducibility**. WGCNA uses individual pairwise correlations for the creation of a co-expression network and gene clustering. We used both the WGCNA gene clusters and the complete gene expression dataset for our covariance analyses. To illustrate how individual pairwise correlations between our genes of interest may be represented in the respective covariance analyses, we performed and visualised a bivariate correlation analysis in R 4.0.3 using the *Hmisc* and *corrplot* packages. All available genes of interest were input into a bivariate correlation analysis. The $p$-value threshold was set to $p < 0.05$. No correction for multiple comparisons was applied as this was used for illustrative purposes.

Partial least squares regression (PLS) analysis was used to identify genes whose expression was most strongly associated with either [$^{11}$C]Ro15-4513 or [$^{11}$C]flumazenil binding. The script used for this analysis is available elsewhere[75] and was run using Matlab R2017a. The approach was similar to that used in previous literature[67,68]. The predictor variable matrix comprised gene expression per brain region in the left hemisphere only since the AHBA only includes data from the right hemisphere for two out of the six donors. The response variable matrices comprised [$^{11}$C]Ro15-4513 and [$^{11}$C]flumazenil binding, respectively, in the 42 brain regions of the left hemisphere. The analysis was then repeated using the 52 WGCNA module eigengenes as the predictor variables. Prior to each PLS analysis, both predictor and response matrices were Z-scored.

The first PLS component (PLS1) is the linear combination of the weighted gene expression scores that have a brain expression map that covaries the most with the map of radiotracer binding. As the components are calculated to explain the maximum covariance between the dependent and independent variables, the first component does not necessarily need to explain the maximum variance in the dependent variable. However, as the number of components calculated increases, they progressively tend to explain less variance in the dependent variable. Here, we tested across a range of components (between 1 and 15) and quantified the relative variance explained by each component (Supplementary Fig. 4). The statistical significance of the variance explained by each component was tested by permuting the response variables 1000 times, while accounting for spatial autocorrelation using a combination of spin rotations for the cortical parcels and random shuffling for the subcortical ones. We decided to focus on the component explaining the largest amount of variance, which in our case was always the first component (PLS1). The error in estimating each gene's PLS1 weight was assessed by bootstrapping, and the ratio of the weight of each gene to its bootstrap standard error was used to calculate the Z-scores and, hence, rank the genes according to their contribution to PLS1. The code used to implement these analyses can be found in https://github.com/SarahMorgan/Morphometric_Similarity_SZ. Result visualisation was performed with ENIGMA toolbox 1.1.1 (https://enigma-toolbox.readthedocs.io/) in Matlab R2018b.

**Reporting summary**. Further information on research design is available in the Nature Research Reporting Summary linked to this article.

## Data availability
The datasets generated during and/or analysed during the current study are provided as Supplementary Data or available in the Figshare repository, https://doi.org/10.6084/m9.figshare.19169663[76]. All other source data can be accessed from the public sources used. Any other data are available from the corresponding author (or other sources, as applicable) on reasonable request.

## Code availability
Code used for any part of the project can be made available at request.

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

## Acknowledgements

This research did not receive any grant from funding agencies in the commercial or not-for-profit sectors. P.B.L. is in receipt of a PhD studentship funded by the National Institute for Health Research (NIHR) Biomedical Research Centre at South London and Maudsley NHS Foundation Trust and King's College London. The views expressed are those of the author(s) and not necessarily those of the NHS, the NIHR or the Department of Health and Social Care. This research was funded in whole, or in part, by the Wellcome Trust [Sir Henry Dale Fellowship 202397/Z/16/Z to G.M.]. For the purpose of open access, the author has applied a CC BY public copyright licence to any Author Accepted Manuscript version arising from this submission. G.M., A.C.V. and F.E.T. acknowledge funding supporting this work from the Medical Research Council UK Centre grant MR/N026063/1. D.M. and V.M. are supported by the National Institute for Health Research (NIHR) Biomedical Research Centre at South London and Maudsley NHS Foundation Trust. The authors would like to extend special thanks to colleagues from the Neuro-biology Research Unit at Copenhagen University Hospital for making their data publicly available, as well as Dr Samuel Cooke for his advice on the interpretation of the results.

## Author contributions

P.B.L., D.M., F.E.T. and G.M. designed the research, P.B.L. conducted the research, D.M. and M.V. provided analytic support, P.B.L. wrote the manuscript and made all figures. A.C.V., P.M. and all other authors edited the manuscript and made contributions to the interpretation of the data.

## Competing interests

The authors declare no competing interests.
