## [Peer Review File · Communications Biology]

Reviewers' comments:

Reviewer #1 (Remarks to the Author):

The manuscript entitled: "Cellular and molecular signatures of in vivo imaging measures of GABAergic neurotransmission in the human brain" represents an innovative and interesting approach for a better understanding of PET molecular imaging. The study is based on the integration of in silico data on regional brain transcripts and GABA receptors evaluated by in vivo using two different radiopharmaceuticals: [C-11]Flumazenil and [C-11]Ro154513. An important point is that the manuscript and in general the method proposed help to better understand which subpopulation of GABAergic interneurons are labeled in vivo by the two radioligands. The manuscript is in general well written and methods and results sections properly described and discussed. Limitations of the study are largely described, however the presence of GABA related markers on astrocytes and the potential impact on these results obtained could be relevant for the readers.

Reviewer #2 (Remarks to the Author):

In this paper the authors attempt to enhance understanding of the molecular origin of GABA receptor PET tracer signal by performing correlation studies using regional PET signal and gene expression levels of GABA receptors and inhibitory interneuron markers. The reported results are novel and highly relevant to provide better insight in the application and interpretation of these GABA receptor PET tracers in clinical populations.

Although there are a few obvious limitations inherent to the approach, which are well summarized in the discussion, the study approach seems valid and described in good detail. The manuscript is very well written, and figures are of high quality.

I have only a few minor questions/comments:

- 1) It is indeed known from in vitro experiments that flumazenil has similar binding affinity to the GABA-A receptors containing different alpha-subunits, while Ro15-4513 is thought to be more selective to GABA-A alpha-5 receptors (10-15-fold). PET signal relates proportionally to affinity and target density, where gene levels might relate mostly to target protein density. Is there literature on the regional protein density of GABRA2 and GABRA3, in comparison to GABRA5, that could help explain their correlation with Ro15-4513 binding?
- 2) There are a few details missing in the section between lines 450-460. Mean and SD of injected radioactivity amount (instead of max) and corresponding data on mass dose should be reported. Line 460 includes "ADD", which should probably be replaced by the appropriate information.
- 3) The PET images of flumazenil and Ro15-4513 were obtained at different centers. Were there any differences, in for example the image processing, that could affect the tracer comparison? This is a potential limitation that could be mentioned in the relevant discussion section.
- 4) It was not clear if corrections for multiple comparisons were applied. Please clarify.

Reviewer #3 (Remarks to the Author):

This publication highlights interesting correlations and anticorrelations of GABAergic gene expression and distribution of well known GABA imaging agents. This is done using a novel way of evaluating the utility and distribution of known PET tracers. The conclusions of distribution of GABA receptor subtypes are in agreement with previous publications, as stated in the manuscript, but the methods (comparing the PET image to the map of gene expression) are novel.

1) Not necessary for this manuscript, but it would be interesting to see these conclusions further explored with the suggested experiments (immunocytochem/autorad/pharm blocking), or to see the gene mapping with some pre-clinical imaging agents targeted at GABA subtypes.

2) The limitation of the size of the AHBA is regrettable as a larger sample size would help to solidify the authors' conclusions. Unfortunately there isn't a feasible work around for this, and the authors did well with the data base as given.

Overall, the authors provide a good contribution to better understanding one of the most important neurotransmitter systems, with a novel approach. They do this while being honest and upfront about the limitations of some of the data (AHBA) and calculations (MRNA doesn't equal gene expression, etc).

Dear Reviewers,

We thank you very much for your insightful comments and suggestions for our manuscript. Our responses to each comment are written in blue text below, and the changes that were made accordingly in the manuscript are made in red both here and in the manuscript.

Reviewer #1 (Remarks to the Author):

The manuscript entitled: “Cellular and molecular signatures of in vivo imaging measures of GABAergic neurotransmission in the human brain” represents an innovative and interesting approach for a better understanding of PET molecular imaging. The study is based on the integration of in silico data on regional brain transcripts and GABAergic neurotransmission evaluated by in vivo using two different radiopharmaceuticals: [¹¹C]Flumazenil and [¹¹C]Ro154513. An important point is that the manuscript and in general the method proposed help to better understand which subpopulation of GABAergic interneurons are labeled in vivo by the two radioligands. The manuscript is in general well written and methods and results sections properly described and discussed.

1. Limitation of the study are largely described, however the presence of GABA related markers on astrocytes and the potential impact on these results obtained could be relevant for the readers.

We thank the Reviewer for this insightful suggestion. Astrocyte markers were included in the dataset we used for cell-type enrichment analyses, results of which can be found in Supplementary Figures 1-3. We initially only described results from these analyses which indicated positive enrichment significant at $p_{FDR} \leq 0.05$. However, we agree that a consideration of the associations between astrocytes and [¹¹C]Ro15-4513/[¹¹C]flumazenil binding may be of interest to our readers. We include the following considerations in our manuscript:

- Results, section ‘[¹¹C]flumazenil binding is associated with *PVALB*, *GABRA1*, *GABRB2*, *GABRG2*, *GABRG3* and *GAD1* expression’, page 13, lines 234-240:

Following up these results with a cell-type enrichment analysis, accounting for weights associated with each gene input into the analysis, revealed enrichment in genes expressed in *PVALB*, *CCK*, *VIP/CCK* and *SST* inhibitory interneurons (Supplementary Fig. 3). The result supported an association between the distribution of *PVALB*, *CCK* and *VIP/CCK* cell-types with [¹¹C]flumazenil binding, and suggested an association between genes enriched in the *SST* cell-type and radiotracer signal despite no direct covariance with *SST* expression found in the PLS analysis. Interestingly, [¹¹C]flumazenil binding was negatively associated with astrocyte-enriched genes.

- Discussion, pages 18-19, lines 321-338:

Interestingly, our cell-type enrichment analysis of radiotracer signal covariance with gene expression showed [¹¹C]flumazenil to be negatively associated with astrocyte-enriched genes, whereas [¹¹C]Ro15-4513 binding did not show such association (Supplementary Fig. 2 and 3). Astrocytes express GABA_ARs⁴⁹, and those containing $\alpha 1$, $\alpha 2$, $\beta 1$ and $\gamma 1$ were shown to be expressed on rat astrocytes⁵⁰. This suggests some GABA PET radiotracer binding may occur on this cell-type. We found [¹¹C]Ro15-4513 and [¹¹C]flumazenil binding to be associated with the expression of genes encoding some of these subunits. We also observed a trend-level enrichment of the astrocyte cell-type in the cluster containing *SST*, which covaried most strongly with [¹¹C]Ro15-4513 signal. Astrocyte distribution is higher in the frontal and temporal cortex⁵¹, where [¹¹C]Ro15-4513 binding is also greater. However, the spatial

associations we set out to identify through our approach relied on the levels of radiotracer binding and gene expression. The degree of GABA_AR expression⁵², as well as *Gabra1* and *Gabra5*⁵³, was shown in rodents to be lower in astrocytes than in neurons. Moreover, the human brain comprises approximately equal numbers of glial and neural cells, and astrocytes represent around 20% of glial cells⁵⁴. This suggests neural GABA_AR expression exceeds that of astrocytes. This may explain the negative covariance between [¹¹C]flumazenil signal and astrocyte-enriched genes, as [¹¹C]flumazenil binding is greatest in posterior brain areas, where astrocyte distribution is lower⁵¹. Correspondingly, it is plausible that the contribution of radiotracer binding to GABA_ARs on astrocytes might be insufficient to considerably drive spatial variation in [¹¹C]Ro15-4513 binding across the brain.

Reviewer #2 (Remarks to the Author):

In this paper the authors attempt to enhance understanding of the molecular origin of GABA receptor PET tracer signal by performing correlation studies using regional PET signal and gene expression levels of GABA receptors and inhibitory interneuron markers. The reported results are novel and highly relevant to provide better insight in the application and interpretation of these GABA receptor PET tracers in clinical populations.

Although there are a few obvious limitations inherent to the approach, which are well summarized in the discussion, the study approach seems valid and described in good detail. The manuscript is very well written, and figures are of high quality.

I have only a few minor questions/comments:

1. It is indeed known from in vitro experiments that flumazenil has similar binding affinity to the GABA-A receptors containing different alpha-subunits, while Ro15-4513 is thought to be more selective to GABA-A alpha-5 receptors (10-15-fold). PET signal relates proportionally to affinity and target density, where gene levels might relate mostly to target protein density. Is there literature on the regional protein density of GABRA2 and GABRA3, in comparison to GABRA5, that could help explain their correlation with Ro15-4513 binding?

Thank you for this insightful comment. We have added the following considerations to the manuscript (pages 16-17, lines 282-301):

It is plausible that the association we found between [¹¹C]Ro15-4513 binding and *CCK*, *GABRA2* and *GABRA3* expression is circumstantial, if enough spatial overlap between the expression of these genes with *SST* and *GABRA5* exists. Both *GABRA2*⁴¹ and *GABRA3*⁴² are highly expressed in the hippocampus, where $\alpha 5$ is most enriched¹⁴. Moreover, *GABRA2* and *GABRA5* follow a similar expression pattern in several brain regions⁴². Alternatively, the finding might reflect secondary affinity of the radiotracer to GABA_AR $\alpha 2/3$. While [¹¹C]Ro15-4513 has 10-15-times greater affinity towards GABA_AR $\alpha 5$ than to GABA_AR $\alpha 1-3$ ²³, it is possible that in regions where GABA_AR $\alpha 5$ is lower, this radiotracer binds secondarily to GABA_AR $\alpha 2/3$. This might deserve further investigation, especially as GABA_AR $\alpha 5$ constitute less than 5% of all GABA_ARs in the brain⁴¹ and PET radiotracers are administered systemically with an intravenous injection, which allows for secondary binding to occur. However, our approach relied on the interrogation of indirect spatial associations between PET radiotracer binding and gene expression across brain regions. As such, direct extrapolations about specific synaptic contributions to our findings warrant future validation with precise molecular methods such as immunocytochemistry or autoradiography with pharmacological blocking. Patterns in regional protein density of specific subunits may help elucidate whether [¹¹C]Ro15-4513 binds to GABA_AR $\alpha 2/3$ in addition to GABA_AR $\alpha 5$. For instance, future studies may assess interlaminar differences in [¹¹C]Ro15-4513 binding, as $\alpha 2/3$ and

$\alpha 5$ present different expression patterns across human prefrontal cortex layers⁴³. In addition, they could investigate the proportion of synaptic to extrasynaptic binding of [¹¹C]Ro15-4513 in human neurons, as GABA_AR $\alpha 5$ are predominantly extrasynaptic¹⁷.

2. There are a few details missing in the section between lines 450-460. Mean and SD of injected radioactivity amount (instead of max) and corresponding data on mass dose should be reported. Line 460 includes "ADD", which should probably be replaced by the appropriate information.

Thank you for pointing this out. We have added the following details to the Methods section of our manuscript (page 25, lines 492-494):

The mean +/- SD amount of radiation administered was 307 MBq +/- 71 (range: 173-405 MBq). PET acquisition was performed in 3D list mode for 70 minutes and binned in the following frames: 15sx10, 60sx3, 120sx5, 300sx11.

3. The PET images of flumazenil and Ro15-4513 were obtained at different centers. Were there any differences, in for example the image processing, that could affect the tracer comparison? This is a potential limitation that could be mentioned in the relevant discussion section.

We thank the Reviewer for raising this issue.

- In the Methods, section 'Parametric map of [¹¹C]flumazenil binding' (page 26, lines 513-517), the following details have been added:

In brief, 16 healthy participants between 16-46 years old (nine females, mean age +/- SD 26.6 +/- 8) were scanned on a CTI/Siemens High-Resolution Research Tomograph. Distribution volume parametric maps were obtained from blood-based Logan graphical analysis, and then normalised to *post-mortem* human brain [³H]diazepam autoradiography data for absolute quantification of benzodiazepine-sensitive receptor density.

- In the Discussion (page 20, lines 380-387), we added the following considerations:

We cannot exclude that the use of two different pipelines for the quantification of [¹¹C]flumazenil and [¹¹C]Ro15-4513 might have had an impact on our results. However, the spatial distribution of radiotracer binding to their target, of importance for this study, has been shown with pharmacological blocking studies to be preserved regardless of which particular quantification method is used⁶⁴. This spatial distribution of signal is also expected to be preserved regardless of the normalisation method originally applied to the [¹¹C]flumazenil map, due to the non-selective nature of binding to benzodiazepine-sensitive receptors of both [¹¹C]flumazenil and diazepam⁶⁴.

4. It was not clear if corrections for multiple comparisons were applied. Please clarify.

We thank the Reviewer for pointing this out and we now clarify this information in the manuscript as follows:

- Methods, section 'Weighted gene co-expression network analysis (WGCNA)' (page 25, lines 473-476):

Gene expression correlation matrix was transformed into an adjacency matrix using the soft threshold power of 14. **This adjacency matrix contained pairwise correlations between all genes in the dataset, uncorrected for multiple comparisons.**

- Methods, section 'Statistics and reproducibility' (pages 26-27, lines 521-527):

WGCNA uses individual pairwise correlations for the creation of a co-expression network and gene clustering. We used both the WGCNA gene clusters and the complete gene expression dataset for our covariance analyses. To illustrate how individual pairwise correlations between our genes of interest may be represented in the respective covariance analyses, we performed and visualised a bivariate correlation analysis in R 4.0.3 using the *Hmisc* and *corrplot* packages. All available genes of interest were input into a bivariate correlation analysis. The p-value threshold was set to $p < 0.05$. **No correction for multiple comparisons was applied as this was used for illustrative purposes.**

Reviewer #3 (Remarks to the Author):

This publication highlights interesting correlations and antecorrelations of GABAergic gene expression and distribution of well known GABA imaging agents. This is done using a novel way of evaluating the utility and distribution of known PET tracers. The conclusions of distribution of GABA receptor subtypes are in agreement with previous publications, as stated in the manuscript, but the methods (comparing the PET image to the map of gene expression) are novel.

Overall, the authors provide a good contribution to better understanding one of the most important neurotransmitter systems, with a novel approach. They do this while being honest and upfront about the limitations of some of the data (AHBA) and calculations (MRNA doesn't equal gene expression, etc).

- 1. Not necessary for this manuscript, but it would be interesting to see these conclusions further explored with the suggested experiments (immunocytochem/autorad/pharm blocking), or to see the gene mapping with some pre-clinical imaging agents targeted at GABAR subtypes.**

We agree with the Reviewer that such future experiments will be an important corroboration of our human findings. However, gathering such data would imply access to samples and methods we unfortunately do not have. Aware of this limitation, we acknowledge this caveat openly in our manuscript (e.g., page 17, lines 292-296) and provide a few suggestions upon which future pharmacological and molecular biology studies can build. As the reviewer acknowledges, it is our opinion that, while interesting, these experiments are not essential to address the questions we set out to investigate in this study. Indeed, our focus at the outset (page 4, lines 82-85 and lines 90-93, page 5, lines 101-104) was to explore these associations in human rather than preclinical data. Moreover, as discussed in the manuscript (page 21, lines 389-392), the scope of this study was primarily to explore the global pattern of covariance between gene expression and radiotracer binding. Where appropriate, we refer to existing preclinical studies (e.g., page 16, lines 274-275, or page 17, lines 304-306) to support our conclusions. Please see below the relevant excerpts from our manuscript:

Discussion, page 17, lines 292-296:

However, our approach relied on the interrogation of indirect spatial associations between PET radiotracer binding and gene expression across brain regions. As such, direct extrapolations about specific synaptic contributions to our findings warrant future validation with precise molecular methods such as immunocytochemistry or autoradiography with pharmacological blocking.

Introduction, page 4, lines 82-85:

Although receptor affinity for these radiotracers has been confirmed in preclinical research²², it is unknown whether this holds inter-species reliability and whether the distribution of distinct cellular and molecular components of the GABAergic system are reflected in radiotracer binding.

Introduction, page 4, lines 90-93:

Moreover, there is abundant rodent evidence of an association between the expression of specific GABA_AR subunits, encoded by individual genes in both presynaptic and target neurons (excitatory or inhibitory), and specific inhibitory interneuron subtypes^{19,24-26}. However, whether analogous organisation patterns are present in human is unclear.

Introduction, page 5, lines 101-104:

Here, we sought to address this issue by using state-of-the-art imaging transcriptomics to 1) uncover patterns of co-expression between genes encoding GABA_AR subunits and inhibitory interneuron markers in the human brain, and to 2) decode their links to the binding distribution patterns of two gold-standard GABA PET radiotracers, [¹¹C]Ro15-4513 and [¹¹C]flumazenil²⁸.

Discussion, page 21, lines 389-392:

Our study did not aim to determine the molecular binding of the two radiotracers, but rather their global pattern of covariance with genes encoding specific GABAergic signalling components. Thus, our findings are correlational by nature and must be interpreted with caution.

Discussion, page 16, lines 274-275:

Preclinical studies have shown that GABA_AR α 5 are enriched on principal cell membranes targeted by SST cells^{5,26,40}.

Discussion, page 17, lines 304-306:

There is prior evidence that [¹⁸F]flumazenil accumulation across the mouse brain after mutations in α 2, α 3 and α 5 subunits, but not in α 1, remained similar to that in wild-type mice⁴⁴.

- 2. The limitation of the size of the AHBA is regrettable as a larger sample size would help to solidify the authors' conclusions. Unfortunately there isn't a feasible work around for this, and the authors did well with the data base as given.**

We thank the Reviewer for this comment and agree with them that this is a limitation of studies using the AHBA dataset, which we acknowledge in the relevant section of our Discussion (page 21, lines 392-396):

Third, the AHBA includes data from six donors only. Samples from the right hemisphere were only collected for two donors, which led us to restrict our analyses to the left hemisphere. Although not a specific limitation of this study, this raises questions about whether this small sample can capture well the principles of organisation of the canonical architecture of gene expression in the human brain and generalise well.

REVIEWERS' COMMENTS:

Reviewer #1 (Remarks to the Author):

I have no other comments

Reviewer #2 (Remarks to the Author):

The authors have answered my questions and I have no further comments.

Reviewer #3 (Remarks to the Author):

This reviewer thanks the authors for the responses submitted to the original comments. Together with the edits made in response to the other reviewers, this manuscript appears ready for publication.